# *onEEGwaveLAD*: A fully automated online EEG wavelet-based learning adaptive denoiser for artefacts identification and mitigation

**Luca Longo**[1,2]*, **Richard B. Reilly**[1]

**1** Trinity Centre for Biomedical Engineering, Trinity College Dublin, Dublin, Ireland, **2** Artificial Intelligence and Cognitive Load Research Lab, Technological University Dublin, Grangegorman, Dublin, Ireland

* luca.longo@tudublin.ie

**Data Availability Statement:** https://erpinfo.org/erp-core.

## Abstract

Electroencephalographic signals are obtained by amplifying and recording the brain's spontaneous biological potential using electrodes positioned on the scalp. While proven to help find changes in brain activity with a high temporal resolution, such signals are contaminated by non-stationary and frequent artefacts. A plethora of noise reduction techniques have been developed, achieving remarkable performance. However, they often require multi-channel information and additional reference signals, are not fully automated, require human intervention and are mostly offline. With the popularity of Brain-Computer Interfaces and the application of Electroencephalography in daily activities and other ecological settings, there is an increasing need for robust, online, near real-time denoising techniques, without additional reference signals, that is fully automated and does not require human supervision nor multi-channel information. This research contributes to the body of knowledge by introducing *onEEGwaveLAD*, a novel, fully automated, ONline, EEG wavelet-based Learning Adaptive Denoiser pipeline for artefact identification and reduction. It is a specific framework that can be instantiated for various types of artefacts paving the path towards real-time denoising. As the first of its kind, it is described and instantiated for the particular problem of blink detection and reduction, and evaluated across a general and a specific analysis of the signal to noise ratio across 30 participants.

## 1 Introduction

Biological measurements, especially those recorded via the Electroencephalography method (EEG), are contaminated by non-stationary and frequent artefacts. These noise sources, usually of an order of magnitude higher than that of a neural signal, contaminate the brain's spontaneous electrical activity. Electrooculographic (EOG) and Electromyographic (EMG) signals are among the most common sources of artefacts in an electroencephalogram because they are rather difficult to prevent. These include eye blinking and twitching of surrounding muscles, which are almost impossible for a human to avoid. This calls for effective noise reduction techniques, which have constantly appeared for the last few decades [1]. While these techniques

**Funding:** The author(s) received no specific funding for this work.

**Competing interests:** NO authors have competing interests.

achieved remarkable performance in denoising EEG signals and improving their signal-to-noise ratio, they often require multi-channel information and an additional reference signal, are not fully automated, require human intervention, and are mostly offline. Given the advent of Brain-computer Interfaces (BCI) and the application of EEG to realise their goals, particularly in daily activities [2] and other ecological, operational settings [3], there is a need for robust, online, near real-time denoising techniques. Unfortunately, the reliability and capability of these techniques in online real-world settings are yet to be established. In particular, the design and deployment of denoising methods and techniques that do not require an additional reference signal, that is fully automated, do not require human supervision, and work on single-channel and in a fully online fashion is still in its infancy [2].

This research focuses on tackling such gap and contributes to the body of knowledge by presenting *onEEGwaveLAD*, a novel, fully automated, online, EEG wavelet-based learning adaptive denoiser for artefact identification and reduction. Specifically, *onEEGwaveLAD* is a framework that can be instantiated for characterising various types of artefacts via wavelet decomposition of single-channel EEG. It is fully automated, with an adaptive mechanism that detects artefacts by learning from a small portion of EEG data preceding the one being recorded. Eventually, it has a strategy to denoise identified artefactual information online. This research study demonstrates how to instantiate and apply such a framework for online blink identification and removal. Fully aware this is only a possible application, as the first of its kind, the long-term goal of *onEEGwaveLAD* is to pave the path towards online denoising EEG signals with a flexible, open and extensible framework.

The remainder of this article is structured over multiple parts. Section 2 motivates the need for such a framework by describing the main sources of artefacts in EEG signals and the properties of existing methods to detect and remove them. Section 3 introduces the *onEEGwave-LAD* solution by presenting its formalisms and their connection. The design of an empirical work based on a particular instantiation of the framework and the description of the research methods are described in section 4. Results are presented in section 5 followed by a critical discussion. Eventually, section 6 summarises this research by describing the open challenges that the presented framework introduced and recommends some specific extensions and evaluation strategies. All these suggestions aim to contribute to the design of robust online denoising methods for electroencephalographic data.

## 2 Related work

Electroencephalography (EEG) is a powerful method for investigating human behaviour. It analyses brain activations recorded by electrodes positioned on the scalp. This method, with a temporal resolution of milliseconds, can record transient brain activations, supporting the identification of anomalous behaviours and the diagnosis of a wide range of neural disorders in clinical practice. Additionally, it is a noninvasive method that is easy to implement. EEG headsets are easy to set up and use, and recent wireless recording equipment makes them portable. Unfortunately, despite these advantages, the temporal signals it can record are contaminated by various artefacts. These can be external, namely extra-physiological, or internal, namely physiological. The recording equipment mainly causes the former, including the movement of the electrodes or cable, and external sources of environmental inferences [4]. The latter is caused by internal sources of psychological activities, including ocular and cardiac artefacts, respiratory activities, skin responses, and muscle contractions. While external artefacts can be prevented by properly setting the equipment, cables, and electrodes, internal artefacts are difficult to mitigate and remove. The reason is that these artefactual sources can lead to signals that highly resemble the recorded EEG signals. In other words, EEG signals are

vulnerable to distortion caused by interfering electrical fields generated by artefactual sources, significantly hampering their signal-to-noise ratio. Identifying and mitigating these interferences has been a challenging problem in neuroscience, with many methods developed in the last few decades.

Generally speaking, according to the additive forward problem in neuroscience [5], cerebral activity $x(n)$, measured via EEG signals, is believed to be a linear combination of electrical waveforms originating from a variety of brain sources $x(n)$, and noise, namely the artefacts $v(n)$ that are instantaneously propagated to the scalp [6]. Formally: $x(n) = s(n) + v(n)$. Consequently, a clear signal can be obtained by subtracting the noise from the recorded cerebral activity. Formally: $s(n) = x(n) - v(n)$. Artefact removal methods can be broadly categorised by their level of human automation and how they operate. In other words, many offline methods have been proposed, some of them requiring human intervention and visual inspection, and some of them fully or partially automated. Among these, regression and filtering methods can be easily implemented. They can function over single or multiple EEG signals with minimal computational complexity, but they require a reference channel or a reliably estimated reference to operate. Examples of regression methods are in [7] while filtering methods are in [8, 9]. Blind source separation approaches, although computationally more complex than the aforementioned methods, can improve denoising performance by estimating artefact-free EEG signals without a reference, but they require multiple channels to function and often human expert intervention for the inspection of components, introducing selection bias in artefact removal [10–12]. Such selection can be made automatic by employing thresholding approaches over the variance of components or their independence. Examples of blind source separation methods include Principal Component Analysis (PCA) [13, 14] and Independent Component Analysis (ICA) [12, 15, 16]. Source decomposition methods represent another class of offline approaches. These can work over a single EEG signal, decomposing it into basic waveforms and leveraging its time and frequency features. They are objective and can be fully automated without human supervision. Still, they require thresholds or prototypical characteristics of an underlying EEG signal and have higher computational complexity than the other offline methods. Examples of source decomposition approaches include the Wavelet Transform (WT) [17, 18], and the Empirical Mode Decomposition (EMD) [19]. Scholars have explored the advantages of multiple techniques and proposed hybrid methods for artefact identification and removal. For instance, spatially constrained ICA and WT have been combined for obtaining full automation [20]. Similarly, source separation was used with EMD and ICA in single-channel recordings [21, 22]. Recent research employing deep learning techniques have showed that denoising can be achieved offline, both semi-automated [23] or fully automated without human supervision [24–26].

Eventually, online methods are rare because they are challenging to implement but probably the most invoked because of the advances in wireless, portable recording equipment for EEG data, enabling real-time monitoring applications, clinical and non. They can work on single-channel EEG data, are fully automated and unsupervised, but have higher computational complexity. In contrast to offline approaches, online methods cannot rely on the data recorded over an entire experiment; they only rely on that up to the current point at which they are employed. In other words, they can rely only on past recorded EEG data at a given time. Online methods are usually divided into block-online and fully online. The former focuses on the analysis and processing of segments of EEG data, usually between 500 and 1000 [27] or even higher [28]. The latter deals with smaller segments, usually between 10 to 50 milliseconds [29]. On the one hand, the main advantage of the block-online method is that since segments are larger, different strategies for denoising can be implemented, especially for longer artefacts such as blinks. However, this requires some delay in delivering the denoised segment for

further use. On the other hand, the fully online method is close to 'real-time'; hence, the processed segment can be delivered almost instantaneously for further use. However, since it processes very short segments, denoising artefacts that last longer than their length and span different consecutive segments is usually difficult.

In synthesis, an online, fully unsupervised, and automated method that does not use reference signals and can be applied to single-channel and multi-channel EEG experiments is lacking. This research study builds on this gap and proposes a real-time framework for solving the aforementioned technical issues, as described in the next section.

## 3 A novel framework for real-time EEG denoising

### 3.1 EEG windowing and sampling rate

The *onEEGwaveLAD* framework's first step is selecting the window length for real-time EEG data collection (Fig 1, phase A). The term 'real-time' here clearly means 'pseudo-real time', the property of a system that receives data, processes it and returns results sufficiently quickly to perform meaningful actions at that time [30]. In other words, the underlying hardware must record a minimum amount of EEG data fast enough that subsequent processing becomes meaningful for a specific purpose [31]. We refer to this parameter as $RTW_L$. The larger the window, the later the post-processing, limiting the real-time effect. Secondly, the sampling rate $S_r$ for data collection of an underlying recording system should be considered. For instance, the larger the sampling rate, the more data is collected in one second; therefore, the higher the quality of EEG data is, and the more granular the post-processing can be. Contrarily, a smaller sampling rate means less data quality and less post-processing effectiveness. The third factor to account for is the upper bound in the frequency domain that a designer/data collector can tolerate. For example, some might be interested in performing post-processing up to $512Hz$, and others only up to gamma frequencies, roughly $80Hz$. Given Nyquist's theorem, a periodic signal must be sampled at over twice its highest frequency component. In other words, given an EEG window length $RTW_L$, and a sampling rate $S_r$, only frequencies up to $((RTW_L * S_r)/1000 * 0.5)$ can be meaningfully extracted from an EEG window. For example, given a $500ms$ window with a sampling rate of $1024Hz$, only $500 * 1024/1000 * 0.5 = 256Hz$ can be meaningfully extracted. The fourth factor is the consideration of the length of certain artefacts. For example,

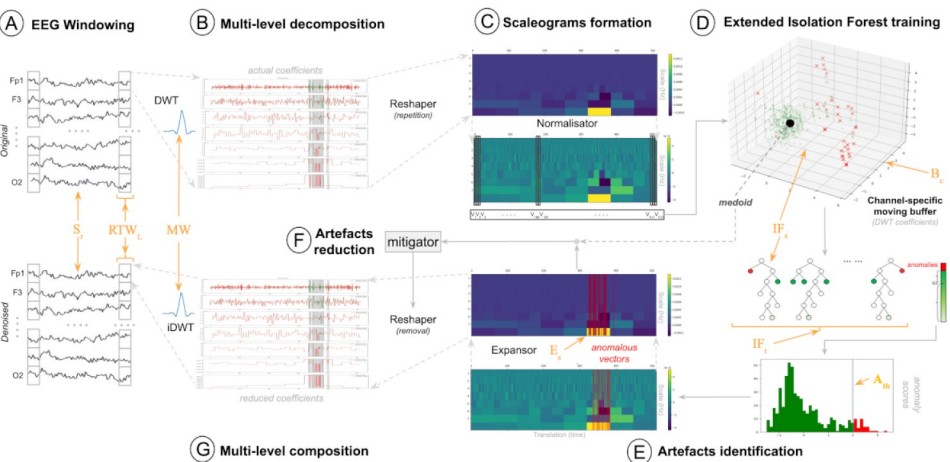

**Fig 1. A schematic representation of the functioning of *onEEGwaveLAD*, an online EEG wavelet-based learning adaptive denoising framework.**

while muscle movements are transient and fast in the time domain, with frequencies ranging within 20–300$Hz$ [32], ocular artefacts are slower in time, with blinks, lasting longer (200ms to 700ms), with frequencies within 4–20$Hz$ [33]. Therefore, if a denoising pipeline is focused on mitigating ocular artefacts in real-time, the window length should be approximately 700$ms$, with a minimum sampling rate of 40$Hz$. Certainly, a blink is not guaranteed to be fully contained in a single window. However, smaller windows might generate problems in identifying blinks. Another technical consideration is that these windows should contain a power of 2 points to allow an efficient decomposition of an underlying signal, as explained in the next section 3.2. The sampling rate, considerations of various artefacts in the time and frequency domains, and the decomposition strategy are interrelated. The *onEEGwaveLAD* pipeline is based on a sliding window technique, whereby consecutive windows of EEG data of length $RTW_L$ are individually collected and submitted sequentially for post-processing, one by one.

## 3.2 DWT multi-level decomposition

Once the window length and sampling rate are set, a single-channel Discrete Wavelet Transform (DWT) is executed. In detail, DWT decomposes a signal into an orthogonal basis obtained by dilation and time translation of a wavelet function $\Psi(t)$, also known as mother wavelet ($MW$), and a scaling function $\Phi(t)$. This process returns a set of coefficients, now referred to as DWT coefficients. This decomposes a signal $x(t)$ of length $n$ into time-frequency components to a level $m \in M$, implemented via a hierarchical set of sub-band filters [34]. DWT was selected over the Continuous Wavelet Transform (CWT) because it can be efficiently implemented with the pyramidal scheme of Mallat [35], known as the sub-band coding scheme, with a computational complexity of $O(|M| * n)$ [34]. Formally:

$$x(t) = \sum_n C_{M,n} \Phi M, n(t) + \sum_{m=1}^{M} \sum_n d_{m,n} \Psi m, n(t) \tag{1}$$

with:

$$\Psi m, n(t) = 2^{-m/2} \Psi(2^{-m} t - n) \tag{2}$$

$$\Phi m, n(t) = 2^{-m/2} \Phi(2^{-m} t - n) \tag{3}$$

where $C_{m,n} = <x, \Phi m, n>$ and $d_{m,n} = <x, \Psi m, n>$ are respectively the approximation and detail coefficients at level $m$. The scalar product is defined with $<f, g> = \sum_{t \in \mathbb{Z}} f(t)g(t)$. As mentioned before, to run DWT, one has to select a mother wavelet $MT$ ($\Psi(t)$). Selecting this function is not a trivial problem in the literature [36]; the nature of an underlying signal and an application domain influences its selection [37, 38].

In summary, the output of the complete DWT transformation of a $x$ univariate EEG signal is a set of approximation and detailed coefficients for each of the various decomposition levels in $M$. These levels do not contain the same amount of coefficients. The pyramidal scheme splits the frequency band into two and applies a wavelet transformation to the highest band. The lower band is then split into two again, and the Wavelet Transform is applied recursively to the higher band of that split. When the output of the low-pass filter cannot be further decomposed, the decomposition scheme terminates. For example, with a window $RTW_L =$ 1000$ms$ (1 second), with a sampling rate of 1024$Hz$, the highest frequency component is 512$Hz$. Therefore, 512 coefficients at level 1 in the [256–512]$Hz$ range of frequencies are produced, 256 at level 2 in the [128–256]$Hz$ are generated and so forth. At the last level of decomposition, only one DWT coefficient is eventually produced, representing the frequencies in the

range $[0.5-1]Hz$. The decomposition scheme produces fewer points at lower frequencies, leading to a decrease in time resolution. In other words, DWT offers poor resolution in the time domain of low frequencies and reasonable resolution in the time domain of high frequencies, providing an effective mechanism for data reduction. It is important to note that the number of decomposition levels depends on the length of the original signal. For a window of length $RTW_L$, then the number of decomposition levels is $log_2 RTW_L$ For the sub-band coding scheme to be efficient, and due to the successive sub-sampling by 2, the signal length must be a power of 2, or at least a multiple of power of 2. For this reason, the window length selection $RTW_L$ should be of the power of 2.

Stretching or shrinking a mother wavelet $MW$ located at a specific point in time leads it to see some or all of an underlying signal. In particular, wavelets applied near the edges of an observed EEG window of length $RTW_L$, inevitably extend their domain outside it. Consequently, the computation of DWT coefficients close to these edges is affected and should be interpreted and used carefully. The extent of the DWT coefficients affected by edge effects depends on the scale used, which is thus closely tied to the frequency under consideration. The higher the scale, the more significant the influence. This phenomenon is often called the cone of influence in the literature [39, 40]. Various techniques have been proposed to compensate for the effect on the DWT coefficients at the boundaries of an observed interval. For example, one focuses on symmetrically reflecting the signal at the border, and another periodically extends it. While interpreting DWT coefficients outside the cone of influence is essential, no formal, precise, agreed approach to determine the cone of influence at each scale exists [41, 42]. In this study, since the denoising pipeline under development has to work in real-time, the problem of the cone of influence is an important one, especially with a continuous stream of recorded EEG windows. While this pipeline can see backwards and consider past recorded data, it cannot see forward since data has not yet been recorded. In the former case, the problem of edge effects on the left border of a current EEG window $w_c$ can be solved by concatenating it to its previous one $w_{c-1}$ and then passing such concatenation ($conc = <w_{c-1}, w_c>$) to the DWT decomposition. In this way, the left border of $w_c$ is now in the middle of *conc*, so it is no longer affected by edge effects. In the latter case, this strategy cannot be applied to the right border of $w_c$ since no window follows it. Therefore, a particular strategy is used, namely a smooth-padding mode, in which an underlying signal is extended according to the first derivatives calculated on the edges (straight line) [43]. This strategy introduces extra DWT coefficients that are eventually trimmed for the subsequent steps but guarantees a smoother DWT decomposition, minimising the edge effects.

## 3.3 Scaleogram formation

As described in the previous phase, the DWT decomposition of a signal occurs at various levels, each with a different amount of approximation and detailed coefficients focused on representing different frequency bands. Consequently, such DWT coefficients cannot be quickly inspected and visually reproduced over time. For this reason, scholars have introduced the notion of 'scaleogram', a visualisation similar to spectrograms but specifically suitable for DWT coefficients. On the one hand, more DWT coefficients exist at lower decomposition levels, representing higher frequencies. On the other hand, at higher decomposition levels, fewer DWT coefficients exist, representing lower frequencies. This asymmetry is resolved by replicating the DWT coefficients at lower levels, twice the amount of the DWT coefficients at the level just above it. In other words, the samples are repeated $2^{m-1}$ times where $m$ is the decomposition level. Consequently, it is possible to have the same amount of DWT coefficients for each decomposition level, forming a matrix. This scaleogram can facilitate visual energy

inspection over time at different frequencies. In detail, the x-axis corresponds to time, and the y-axis corresponds to the decomposition scales, the levels (Fig 1, phase C, top scaleogram). The scale is the signal periodicity to which the transform is sensitive, and each value in the scaleogram represents the amplitude of the signal variation measured. In other words, these variations are located at specific times with specific periodicity. Unfortunately, the DWT coefficients at higher levels (lower frequencies) are generally more significant in magnitude than those at lower levels (high frequencies). Therefore, their impact is different if used in subsequent computations, with higher-level coefficients being more critical. A normalisation strategy is proposed to counteract this and assign the same importance to coefficients at the different decomposition levels (the scales). This divides each coefficient $d$ at level $m$ by $2^m$ with $m$ starting at 1.

$$d_{norm} = \frac{d}{2^m}$$

with $d_{norm}$ the normalised DWT coefficient at level $m$. The normalised scaleogram is a matrix $n \times m$, with $n$ the amount of normalised DWT coefficients per each of the $m$ decomposition levels. $n$ equates to the maximum frequencies that can be extracted from a signal of length $RTW_L$ and sampling rate $S_r$. As mentioned in section 3.1, and according to Nyquist's rule:

$$n = \frac{(RTW_L * S_r)}{1000} * \frac{1}{2}$$

with $RTW_L$ in milliseconds, and $S_r$ in hertz.

## 3.4 Extended isolation forest and adaptive moving buffer

Each slice of the $n \times m$ matrix, as produced in the previous step, is a $n$-dimensional vector, a unique representation of the signal at a given point in time in the domain of DWT coefficients. The first assumption is that those vectors associated with artefacts in the underlying EEG signal they represent are rare and different from all the others. Thus, they can be considered anomalies. In general, many approaches have been devised to efficiently identify anomalies within a group of points in the one-dimensional space. Among these popular solutions are those based on statistics, classification-based, clustering-based, Nearest Neighbor-based, Information-theoretic and spectral techniques [44]. Unfortunately, while they can achieve remarkable accuracy in detecting anomalies, and with a linear time complexity in testing them, they are often computationally expensive in time and space during training, often with quadratic complexities [44]. The training phase is thus a bottleneck for developing real-time anomaly detection applications. Instead, a novel algorithm, the Isolation Forest (iForest), has a linear time complexity and a limited requirement for memory consumption [45]. This makes it an appealing candidate solution for developing a real-time EEG denoiser. The main advantages of this approach over the existing ones are multiple. Firstly, it is model-free, and no statistics or probabilities, including density estimation or statistics on the class distribution, are required. Secondly, it is fully unsupervised and does not require labelled ground truth. Thirdly, it has been proven effective on n-dimensional data, and, through a sub-sampling procedure, it effectively solves the problems of swamping, which means the vicinity of normal points to the anomalous one, and that of masking, when many anomalies exist [46]. The assumption behind this algorithm is that anomalies within data are specific points that are limited in cardinality and vastly different from most of the other points, thus making them easier to separate. In other words, anomalies are rare instances of a dataset, and their characteristics in an n-

dimensional space should be drastically different from those of the other instances and, therefore, easily identifiable.

Given a dataset $X = x_1, \ldots, x_c$, where each instance is of dimension $c$, iForest sum-samples a random portion of data $X' < X$ to construct a binary tree, namely an Isolation Tree (iTree). The branching process of such a tree is executed by picking a random dimension $rd_i$, with $i$ in $1, 2, \ldots, c$, and a random split value $sv$, within the range of $rd_i$ ($sv \in [min(rd_i), max(rd_i)]$). If the dimension $rd_i$ of a particular instance of the dataset (data point) has a value smaller than $sv$, then such instance is assigned to the left branch of the iTree. In contrast, the instance is assigned to its right branch, splitting the data on the current node of the tree into two. This branching mechanism is then performed recursively for each instance of the dataset until a single point is isolated or a predetermined depth limit is reached. In other words, until the node has only one instance, or all the data at that node have the same values. Formally, an iTree is a data structure where each node $T$ is either a leaf, an external node with no child, or an internal node with exactly two children nodes ($T_l$, $T_r$), and a condition defined by a dimension $rd_i$, and a split value $sv$, such that $rd_i < sv$ determines the traversal of a data point to the left branch $t_l$, or the right branch $T_r$. When the iTree is fully grown, each point in the dataset $X$ is isolated at one of the leaf nodes. Each data point $x_i \in X$ has a specific path length $h(x_i)$, which equates to the number of edges $x_i$ traverses from the root to a leaf in the tree. Intuitively, an anomaly is a data point with a smaller path length since it can be isolated easily. The aforementioned recursive process called the training phase, is repeated $IF_t$ times, creating an ensemble of iTrees, an isolation forest (iF). The identification of anomalies can be performed by a binary search with such a forest. In detail, similarly to the searching mechanism of the Binary Search Trees (BST) algorithm [47], the termination to a leaf node on an iTree means an unsuccessful search in the BST. Consequently, estimating the average path length $h(x)$ of all the leaf nodes is equivalent to the unsuccessful search in the BST. Formally:

$$a(|X'|) = \begin{cases} 2H(|X'| - 1) - \dfrac{2(|X'| - 1)}{|Y|} & \text{for } |X'| > 2 \\ 1 & \text{for } |X'| = 2 \\ 0 & \text{otherwise} \end{cases} \tag{4}$$

where $|Y|$ is the size of the testing data $Y$, $|X'|$ is the size of the random portion of data $X'$, $H$ is the harmonic number that can be estimated by $H(i) = ln(i) + \gamma$ (with $\gamma = 0.5772156649$ the Euler-Mascheroni constant). In other words, $a(|X'|)$ is the average of $h(x)$ given $|X'|$, and it can be used to normalise $h(x)$ and obtain an estimation of the anomaly score for a specific data instance $x \in X$. Formally:

$$s(x, a(|X'|)) = 2^{\frac{-E(h(x))}{a(a(|X'|))}} \tag{5}$$

with $E(h(x))$ the average value of $h(x)$ from the forest, the collection of iTrees. For any given instance $x \in X$, if $s$ is smaller than 0.5, then $x$ is likely to be a normal value (not an anomaly). In contrast, if $s$ is close to 1, the instance $x$ is likely to be an anomaly. Once an isolation forest is built, a test phase is executed whereby each instance of a test set $Y = y_1, \ldots, y_c$ is passed to each iTree of the forest, and an anomaly score is assigned to it. In this research, a particular version of the original Isolation Forest is used, namely the Extended Isolation Forest algorithm [48]. Such an extension solves the problem of bias in the original algorithm caused by tree branching. In detail, a feature and a value are chosen at each branching point, which is a form of bias since such a point is parallel to one of the axes. In contrast, the extended algorithm defines a random slope $rs$ instead of selecting a feature and value for the branching cut and a random

intercept *ri*. On the one hand, a Gaussian distribution $\mathcal{N}(0, 1)$ can generate such a slope *rs*. On the other hand, the intercept *ri* can be generated from the uniform distribution with bounds coming from the sub-sample of data that has to be split. Formally, the branching criteria for the data splitting for a given point is $(x - ri) * \leq rs$. One parameter of the Extended Isolation Forest is the extension level *el*, which is a number in the range $[0, c - 1]$, with *c* the number of features (dimension) of an original instance $x \in X$. If *el* = 0, the extended version coincides with the original Isolation Forest Algorithm. Deviations from zero correspond to reductions of bias. Another parameter is the number of randomly sampled observations $IF_S$ used to train each Extended Isolation Forest tree.

The above mechanism can identify the anomalous $n-$dimensional vectors among the *m* vectors of each normalised scaleogram, if any, as produced in the previous step for each EEG window. Ideally, more of these vectors are used as input to the Isolation Forest algorithm, and better anomaly detection can be performed. Theoretically, suppose such an algorithm is executed after collecting an EEG window of length $RTW_L$. Consider also that an isolation forest *iF* was trained with all vectors composing each normalised scaleogram of its preceding EEG windows. In such case, if an *iF* is trained after collecting each EEG window, increasing training data is produced over time; thus, a longer time is required to train it. However, while it is theoretically possible to include all the vectors of each normalised scaleogram collected up to a point in time, practically training an Isolation Forest with all such vectors requires a linear increment in time complexity, proportional to the number of recorder EEG windows. Consequently, testing an *iF* at a future point might cause technical issues because it might require more time than the established length of the EEG window ($RTW_L$), effectively hampering the effort to develop a real-time EEG denoiser. In other words, accounting for all the previous vectors of normalised scaleograms recorded up to a point in time is not ideal if a real-time denoising pipeline for artefacts in EEG signals is to be devised. For this reason, a moving, sliding buffer $B_s$ of fixed capacity *s* is proposed to contain the *n*-dimensional vectors of the normalised scaleograms associated with the *s* recorded EEG windows occurring earlier than the one recorded in real-time. On the one hand, this fixed capacity resolves the technical challenges associated with the time complexity required to run the Isolation Forest algorithm and test the resulting model. On the other hand, the buffer's 'sliding' property also makes the proposed artefact denoising pipeline adaptive. It is adaptive to the changing recording environment and inherently prone to the slow amplitude drifts of each EEG channel. These might be caused by the reduction of the scalp electrodes' physical adherence or the drying of the conductive gel used, among other reasons. However, such consideration entails a second assumption behind this research study: genuine neural signals are recorded in most parts of this moving buffer, with a small portion of artefacts. While this is reasonable, identifying anomalous EEG segments will be hampered if the buffer contains mainly noise and artefacts. The investigation of this assumption is outside the scope of this study and will be the subject of future empirical investigations. Intuitively, solutions can be defined to check whether the sliding buffer mainly contains neural plausible signals.

### 3.5 Artefact identification

Once the extended Isolation Forest algorithm is run with the *n*-dimensional vectors of all the scaleograms saved in the moving buffer up to a time point for a given EEG channel, it produces a list of anomaly scores, one score for each vector. Subsequently, an anomaly threshold $t_a$ is devised to discriminate abnormal $n-$dimensional vectors from the rest. As mentioned at the end of section 3.4, a rule of thumb suggests that a value of $t_a$ = 0.5 can be confidently used to separate the $n-$dimensional vectors that behave normally from those that do not [45]. When

the extended Isolation Forest algorithm is run with the $n$-dimensional vectors in the moving buffer, hereby referred to as the 'training' procedure, producing an Isolation Forest $iF$, it is then tested with the $n$-dimensional vectors associated with the normalised scaleogram of the current EEG window recorded in real-time (the $B_s$ + 1th EEG window, with $B_s$ the size of the moving buffer). This 'testing' procedure allows computing an anomaly score for each of the $m$ $n$-dimensional vectors of the $B_s$ + 1 normalised scaleogram (the current one) and extracting those with a value greater than the threshold $t_a$, if any. The timestamp of each of these abnormal vectors is saved in a list $AT$ (anomalous timestamps). In other words, these timestamps, if any, represent the exact points in time within the current EEG window that contain abnormal behaviour (the anomalous vectors) and potential artefacts.

## 3.6 Artefact reduction

Once the abnormal time locations, the list $AT$ (anomalous vectors, in red in Fig 1) have been established, if any, for the $n-$dimensional vectors in the current EEG window, their deviations from normal behaviour, the potential artefacts, need to be mitigated. At this stage, the notion of a normalised scaleogram is forgotten, and the original, non-normalised scaleogram of the current EEG window is instead re-considered. This is because, eventually, the denoised pipeline proposed in this research will culminate in the output of a neural signal in the time domain by applying the inverse Discrete Wavelet Transform (DWT) on the denoised vectors for the current EEG window. This inversion requires a set of concatenated approximate and detailed DWT coefficients to generate a time series. If such coefficients are those normalised across levels (as described in section 3.3), then this inversion would not produce a denoised signal close to the original one but a different one, thus essentially and erroneously changing the underlying neural dynamics. For this reason, the non-normalised (original) DWT coefficients of the current EEG windows are considered and manipulated before being inverted into the time domain.

Artefact reduction usually includes removing or attenuating wavelet coefficients using a thresholding mechanism [49]. Technically, a threshold could be established by visually inspecting the wavelet coefficients. However, while this works in offline artefact reduction, it is unfeasible in online applications. Therefore, a common approach is to use the Universal Threshold considering all the wavelet coefficients and their standard deviation [50]. Subsequently, artefact reduction is usually implemented by applying a hard or a soft thresholding function on the wavelet coefficients [51]. Threshold values can be generated by considering all the wavelet coefficients for all the DWT levels or separately for each of them [52]. The latter case is relevant in EEG artefact removal applications where stationarity is not guaranteed [53]. For this reason, other approaches have been proposed to account for deviations from stationarity, where time-varying thresholds are defined by considering the statistical deviation computed over an ensemble of surrogate time series [54].

Inspired by this property of adaptivity, *onEEGwaveLAD*, via its sliding buffer, proposes an adaptive thresholding mechanism for DWT coefficients at the different DWT levels. This mechanism includes computing the medoid $B_{medoid}$ of all the $n-$dimensional vectors in such moving buffer as the most representative point, whose sum of dissimilarities to all the other vectors is minimal. It is considered the most common behaviour of an underlying neural signal in the preceding EEG windows to the current one (with $B_p$, the number of windows in the buffer), including information about its frequencies over time. Additionally, motivated by wavelet theory [55] and the fact that the wavelet coefficients around a vector identified as anomalous are similar in magnitude, an expansion step $E_s$ is introduced, defining how many neighboured vectors to those identified as anomalous should also be denoised. This is

important because some neighbouring vectors might not have been identified as anomalies because they do not satisfy the anomaly threshold $t_a$. This expansion leads to a new list of time-stamps $AT_{exp}$ of anomalous vectors. Such a denoising mechanism aims to implement a smooth artefact reduction around each anomalous vector, not a sharp one. It also considers the neural mechanisms detected in the EEG data before the window is recorded. In detail, the denoising of the wavelet coefficients of the $n-$dimensional vectors in the expanded list $AT_{exp}$ can be implemented via a mitigator vector $Mtg$, which is the complement of the distance of the specific DWT coefficients of a vector $a$ in the list $AT_{exp}$, from the medoid $B_{medoid}$ of the buffer $B_s$, divided by the max distance vector, always from the medoid, among the vectors in the buffer $B_s$. Formally:

$$\overrightarrow{mtg_a} = 1 - \frac{d(\overrightarrow{a}, B_{medoid})}{max(d(\overrightarrow{v}, B_{medoid}) : \ \forall v \in B_s)} \quad \forall a \in AT_{exp} \tag{6}$$

with $\overrightarrow{mtg_a} : [0..1] \in \Re$ is the mitigating vector used to denoise every vector $a$ in $AT_{exp}$. This mitigator is more aggressive for vectors far from the medoid (the potential artefacts) and softer for those close to it (the normal neural signal). The new vector $\overrightarrow{a_{den}}$ of denoised DWT coefficients of each of the $n-$dimensional vectors in the expanded list $AT_{exp}$ is its Hadamard Product, namely the element-wise multiplication, with the mitigator vector. Formally:

$$\overrightarrow{a_{den}} = \overrightarrow{a} \odot \overrightarrow{mtg_a} \tag{7}$$

It is important to observe that even if an $n-$dimensional vector is identified as anomalous, and because its denoised version is computed element-wise, some of its DWT coefficients can be reduced more aggressively, for example, at some specific scale (frequency) than others. Future extensions of this might include a mechanism that mitigate only certain frequencies known to be important for a particular artefact, for example such as blinks.

## 3.7 Multi-level composition

Eventually, the last step in the overall denoising pipeline is the concatenation of the non-normalised $n-$dimensional vectors with those that went through the artefact reduction phase, as described in the previous section 3.6, preserving their time appearance. This concatenation represents the denoised scaleogram for an underlying length signal $n$. At this stage, the DWT coefficients in this scaleogram have been repeated $2^{m-1}$ times for each decomposition level $m$, as already described in section 3.3. To properly re-convert them into the time domain, they must be reshaped, and the repetitions removed (reshaper-removal in Fig 1). Specifically, for the first decomposition level, with a window of length $RTW_L$, with a sampling rate $S_r$ in hertz, $k = ((RTW_L * S_r)/1000 * 0.5)$ DWT coefficient exists. Consequently, for each decomposition level $m$, only $k/2^{m-1}$ coefficients are unique and not repeated; the other must be removed. This can be done by taking the first DWT coefficient for every $2^{m-1}$ coefficient at a level $m$. This reshaping procedure results in more DWT coefficients at lower decomposition levels, representing higher frequencies, and fewer at higher levels, representing lower frequencies. In particular, this has the same shape as the output of Eq 1, with denoised approximate coefficients $C_{M,n}^{den}$ for a signal of length $n$ and denoised detailed coefficients $d_{m,n}^{den}$ for each decomposition level $m \in M$. These are subsequently passed through the inverse Discrete Wavelet Transformation (DWT) (inversion of Eq 1) for their conversion back into the time domain. Formally:

$$x^{den}(t) = \sum_n C_{M,n}^{den} \Phi M, n(t) + \sum_{m=1}^{M} \sum_n d_{m,n}^{den} \Psi m, n(t) \tag{8}$$

where $C_{M,n}^{den} = <x, \Phi M, n>$ and $d_{m,n}^{den} = <x, \Psi m, n>$ are respectively the sets of denoised approximation and detail coefficients. This eventually returns a corrected signal $x^{den}(t)$, which is the denoised version of the original signal over time.

### 3.8 Summary of parameters of *onEEGwaveLAD*

In synthesis, the *onEEGwaveLAD* pipeline has a set of parameters dependent on the type of artefact that needs to be removed, the nature of the recorded EEG signal, and the strategies used to detect and reduce noise in it:

- *Real-time EEG Window Length* ($RTW_L$): the length in milliseconds of a recorded EEG segment for dealing with real-time denoising;

- *Sampling rate* ($S_r$): the number of points, in Hertz, of the recorded EEG segment, for dealing with the granularity of denoising;

- *Mother wavelet*($MW$): a function used for decomposing the recorded EEG segment, employing the DWT decomposition scheme; this function should be as similar as possible to the prototypical shape of the artefact being removed;

- *Buffer capacity* ($B_s$): the amount of EEG windows composing the sliding buffer for storing the past EEG signal's behaviour to the current recorded window; this is for implementing the adaptiveness of the denoising framework to the non-stationarity of the recorded EEG signal;

- *IF Sub-sampling size* ($IF_S$): the number of randomly sampled observations used to train each Extended Isolation Forest tree;

- *Number of IF trees* ($IF_t$): the number of trees to learn by the Extended Isolation Forest Algorithm;

- *Anomaly threshold* ($t_a$): a threshold for deeming a vector of $n$-dimensions (the decomposition scales) as an outlier given its anomaly score computed by the extended Isolation Forest model;

- *Expansion step* ($E_s$): the number of points to consider around each anomalous vector that has to be denoised.

## 4 Design and methodology

An experiment is designed to instantiate *onEEGwaveLAD*, the real-time denoising pipeline and assess it for the first time. In particular, such a pipeline is tested for identifying and removing blinks, one of the most frequent types of artefact present in EEG signals. Fig 2 summarises the components of such experimental design, and the following sections describe each in detail.

### 4.1 Data understanding

The N170 dataset of the ERP-CORE (Compendium of Open Resources and Experiments) was selected (https://doi.org/10.18115/D5JW4R). It includes EEG data from 40 participants (25 female, 15 male; mean years of age = 21.5, SD = 2.87, Range 18–30; 38 right-handed) from the University of California, Davis community who were exposed to a visual discrimination paradigm for isolating the face-specific N170 response. Each participant reported their native

**Fig 2. High-level diagram of the design of an experiment for evaluating *onEEGwaveLAD*: A fully automated online EEG wavelet-based learning adaptive denoiser for artefact identification and mitigation.**

English competence, normal colour perception, normal or corrected-to-normal vision, and no neurological injury or disease history. As suggested in [56], participants who exhibited a significant portion of artefacts from a visual inspection were removed. This led to a final dataset containing EEG data from 37 participants, and only raw data was considered for this experiment. The recording length averages 581 seconds with a standard deviation of 55 seconds across participants. EEG data was recorded using a Biosemi ActiveTwo recording system with active electrodes (Biosemi B.V., Amsterdam, the Netherlands) from 30 scalp electrodes, placed according to the 10/20 International System (FP1, F3, F7, FC3, C3, C5, P3, P7, P9, PO7, PO3, O1, Oz, Pz, CPz, FP2, Fz, F4, F8, FC4, FCz, Cz, C4, C6, P4, P8, P10, PO8, PO4, O2). P01 was assigned to the common mode sense (CMS) electrode, with the driven right leg (DRL) electrode at PO2. Signals were low-pass filtered using a fifth-order sinc filter with a half-power cut-off at 204.8 Hz and then digitized at 1024 Hz with 24 bits of resolution. Electrodes were placed lateral to the external canthus of each eye to record the horizontal electrooculogram (HEOG). In addition, a vertical electrooculogram (VEOG) was recorded using an electrode placed below the right eye. Further details on how data was collected and the original context of application to Event-Related Potential research can be found in [57]. The primary rationale for selecting such a dataset is the public availability of the ERP-CORE. This was prepared by a well-known research group in neuroscience, which suggested that researchers could use it to test new hypotheses by reanalyzing the existing data in novel ways and to test newly developed data processing procedures, as is eventually done in this research study.

### 4.2 Instantiation of *onEEGwaveLAD*

The *onEEGwaveLAD* pipeline described in section 3 is instantiated for ocular artefact identification and reduction, specifically for blinks (and not saccades) with the parameters listed in Table 1.

The Real-time EEG Window Length of 1 second is selected because of the typical blink length: between 100 and 400ms [58]. In the best case, the rationale is to have a length long enough to include the entire blink duration. The sampling rate was kept at the original, 1024Hz. Since this is a multiple of a power of 2, this aligns with the suggestions related to the efficiency of the sub-band coding scheme of DWT multi-level decomposition, as described in section 3.2. The Mother Wavelet selected is 'Sym4', a near symmetric (least asymmetric), orthogonal and biorthogonal function of the Daubechies family of wavelets, already used in experiments dealing with ocular artefacts [59, 60]. This is chosen because its shape highly resembles an ocular eye blink [61] (Fig 2, middle) with a positive peak. The buffer capacity is chosen to be 10, 15, and 20 windows, which, given their length of one second, correspond to 10, 15, and 20 seconds, respectively. Three options are tested to evaluate their impact on the

**Table 1. Parameters of an instantiation of the *onEEGwaveLAD* pipeline for blink identification and reduction for the Fp1 and Fp2 EEG channels.**

| Description | Parameter | value |
|---|---|---|
| *Real-time EEG Window Length* | $RTW_L$ | 1000 ms |
| *Sampling rate* | $S_r$ | 1024 Hz |
| *Mother wavelet* | $MW$ | Sym4 |
| *Buffer capacity* | $B_s$ | [10, 15, 20] windows |
| *IF sub-sampling size* | $IF_S$ | 512 samples |
| *Number of IF trees* | $IF_t$ | 100 trees |
| *Anomaly threshold* | $t_a$ | [0.55] |
| *Expansion step* | $E_s$ | [0, 5, 10, 15, 20, 25, 30, 35] |

overall denoising capacity of the proposed pipeline, keeping the sub-sampling size invariant. The Isolation Forest algorithm's sub-sampling size is set to 512, which equates to the number of DWT coefficients in a single EEG window given a sampling rate of 1024. Considering the buffer capacity options mentioned before, the Isolation Forest sub-samples have 512 DWT coefficients, respectively, out of 5120, 7680, and 10240, which are the number of DWT coefficients in the three buffer capacities (10%, 6%, 5%). Such sizes are believed to help tackle the problems of swamping and masking when detecting anomalies using the Isolation Forest algorithm. The number of trees chosen as a parameter for such an algorithm is 100, a reasonable number to learn a robust forest for discriminating anomalies. The anomaly threshold to consider a $n-$dimensional vector of DWT coefficients as an anomaly is 0.55. This is slightly more stringent than the rule of thumb described in section 3.5. In other words, if the anomaly score, computed by using the Isolation Forest model, of each vector composing the normalised scaleogram of the current EEG window (the last recorded), trained with all the vectors associated with the normalised scaleograms of the previous EEG windows saved in the moving buffer, is above 0.55, then it is considered an anomaly, a specific outlier. Eventually, the expansion step is tested against eight options: 0, 5, 10, 15, 20, 25, 30, 35. Once anomalous DWT coefficients in the current EEG window are found, if any, an expansion step of 35 means that 35 points to their left and 35 points to their right are marked as DWT coefficients that also require reduction. If such expansion exceeds the boundaries of the current EEG windows, they are clipped (0 to the left side and 512 to the right side). In other words, the expansion does not exceed the boundary of the current EEG window (containing 512 coefficients). These options are tested to investigate the impact of such expansion on the denoising capacity of the proposed pipeline. As mentioned in section 3.6, the reasoning is that artefacts are not only on the DWT coefficients deemed anomalous but, given their transient nature, also in their neighbourhood.

## 4.3 Evaluation

A ground truth is required to test the capacity of *onEEGwaveLAD* for detecting and removing blinks with the specific instantiation described in the previous section. Identifying blinks via an offline analysis of the entirety of a recorded EEG signal has been proven not to be a complicated problem [62, 63]. This can be done with a peak detection algorithm on specific uni-variate signals, such as that recorded by a vertical EOG electrode (VEOG), if present [64]. Alternatively, in its absence, Fp1 and/or Fp2 can be used [65]. Unfortunately, peak detection algorithms suffer from the problem of manually setting a threshold that can be used to discriminate blinks in a uni-variate signal [66]. Lower thresholds might lead to identifying few blinks, decreasing true positives, while higher thresholds can increase false positives. In this

experiment, the VEOG channel is selected, given its availability in the chosen dataset (ERP-CORE/N170), and a peak detection algorithm (algorithm 1) is adopted to extract a list of potential peaks.

**Algorithm 1** Peak detection algorithm

```
procedure PEAKDETECTION[VEOGsig], k
    [VEOGsig_fir] ← firwin([VEOGsig], 1, 10)
    [VEOGsig_abs] ← abs([VEOGsig])
    [VEOGsig_sqrt] ← sqrt([VEOGsig_abs])
    VEOGsig_max ← max([VEOGsig_sqrt])
    VEOGsig_std ← std([VEOGsig_sqrt])
    VEOGsig_th ← (VEOGsig_max - VEOGsig_std)/k
    peaks ← findPeaks(VEOGsig_fil, VEOGsig_th)
    return [peaks]                                  ▷ Peaks locations
end procedure
```

A filtered VEOG signal, achieved by applying a Firwin filter in the range [1, 10]Hz, is converted to absolute values and root squared. The resulting signal's max and standard deviation are computed, and the threshold is established with a tunable $k$ value passed as input. Tuning this value can result in more blinks detected, which can be detrimental to false positives, or fewer blinks, which can be detrimental to true positives. Eventually, a function for finding peaks given the filtered VEOG and the established threshold is run, and a list of peak locations is computed. Many of these functions exist in the literature, and one can select the most appropriate. Such a peak detection algorithm is subsequently invoked with different increasing $k$ values to find true positives and a particular multi-channel selection strategy to filter false positives, as in the pseudo-code in S1 Appendix 7.1. In particular, it is known that a blink has a positive charge in the cornea, close to the pre-frontal channels, especially Fp1 and Fp2, and a negative charge in the retina, in the occipital channels, especially in O1, O2, Oz [67, 68]. Frontal channels (F) are affected by the positive charge, less than the pre-frontal ones (Fp). The parietal occipital channels (PO) are affected by the negative charge, but higher than the occipital ones (O). A loop goes through various options for parameter $k$ and either stops at the last option, or when the number of identified true positive blinks does not improve for some consecutive times (patience). The topographic maps associated with these peaks, computed using the multi-channel information at that given time, are manually visually inspected for a final confirmation, and their timestamps are saved in a list of true blinks.

For each EEG window, the signal-to-noise ratio (SNR) is computed for the original (raw) signal and its denoised version for the channels Fp1 and Fp2. A denoised EEG window is considered a true positive if *onEEGwaveLAD* had identified at least one anomalous $n-$dimensional vector in its DWT space. The hypothesis is that the *onEEGwaveLAD* real-time denoising pipeline is expected to have better SNR ratios for the true positive blinks but similar ratios for the true negatives. In the non-blink EEG windows, the SNR will be the same for false positives since no artefact will be identified, and thus, no reduction of their EEG signals will be performed. Similarly, the false negative EEG windows will be modified since the *onEEGwaveLAD* will erroneously identify some parts as anomalous. However, in this case, the SNR in the false negative EEG windows is expected to be lower than in the true positive regions. For this, the Jensen-Shannon (JS) divergence between the probability distributions of the original signal in the true positive EEG windows and the denoised signal in the same windows is expected to be higher than that in the false negative EEG windows. The JS metric, grounded on information theory, returns a value between zero and one, with zero indicating that the two underlying probability distributions are identical, and one determines that they are entirely different. It was adopted because it offers a way of calculating the drift between original and denoised EEG signals. It is a symmetric metric, meaning that the order of the two underlying distributions

does not impact the resulting divergence computation. Similarly to the Kullback-Leibler (KL) divergence, it can be thought of as a measure of the distance between two data distributions aimed at showing how they differ from each other. However, in contrast to the KL divergence, a mixture probability distribution derived from the two underlying distributions is used as a baseline when comparing them instead of a static distribution that needs to be assumed before computation.

In synthesis, the goal is to demonstrate that *onEEGwaveLAD* can effectively denoise artefactual intervals but not alter non-artefactual intervals, thus changing genuine neural activity.

## 5 Results and discussion

Following the previous design, an instantiation of the *onEEGwaveLAD* pipeline with the parameter of Table 1 was executed with the EEG data from the selected 30 subjects. Fig 3 depicts an example of the DWT coefficients obtained by applying *onEEGwaveLAD* on an EEG window 1-second long, at 1024 Hz, at the eight scales. As formally described in section 3.3, the first scale contains an amount of DWT coefficients, in this case, 512, which is half of the sampling rate. At level 2, this amount further shrinks by half, and so on until the 8th scale. Fig 6 in S1 Appendix depicts the formation of the ground truth, which means the selection of plausible blinks with algorithm 2 S1 Appendix out of all the peaks found with the peak detection algorithm 1 (section 4.3).

Fig 4 introduces the density plots of the accuracies obtained. The two columns of plots depict results associated with the frontal Fp1 and Fp2 Channels, respectively, while the rows depict the results obtained by varying the buffer size ($B_s$=[10, 15, 20]). Each plot contains eight distributions, one for each expansion step ($E_s$=[0, 5, 10, 15, 20, 25, 30, 35]). The computation of the accuracy is done as:

$$Accuracy = \frac{TP + TN}{TP + TN + FP + FN}$$

where TP is the true positives (blinks correctly predicted), TN is the true negatives (non-blinks correctly predicted), FP is the false positives (blinks erroneously predicted as non-blink), and the false negatives (non-blinks erroneously predicted as blinks).

The first noticeable finding is that accuracies are around 0.5 in all the cases, demonstrating an initial reasonable capability to spot blinks and non-blinks in real time. While the reader

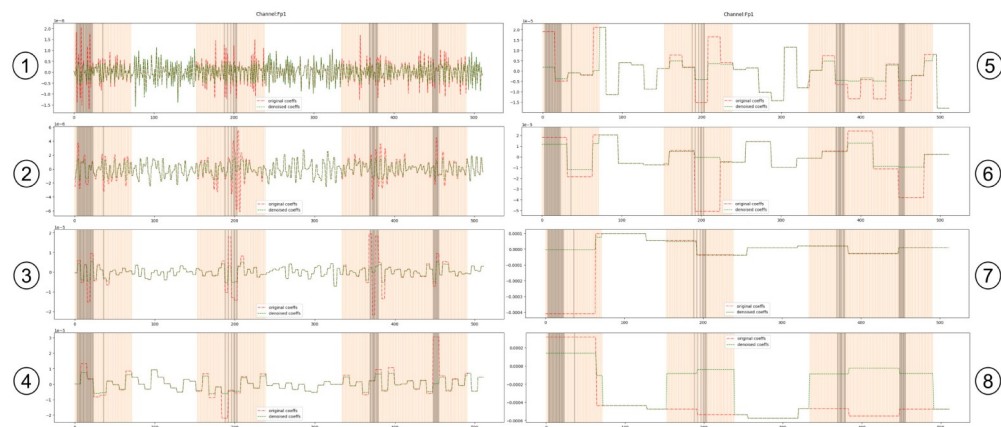

**Fig 3. An illustration of coefficient space over eight scales resulting from applying the Discrete Wavelet Transform of a neural signal.**

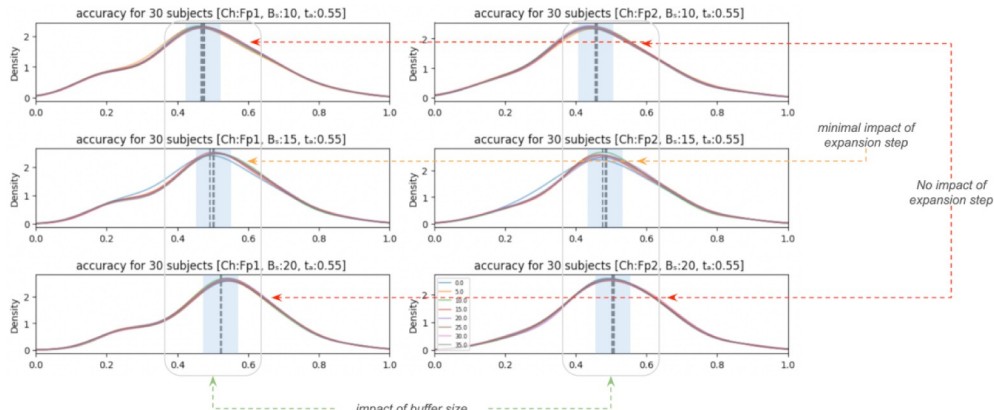

**Fig 4. Distributions of the accuracies for blink identification by the onEEGwaveLAD at each expansion step, grouped by buffer size (column) and the pre-frontal channels (Fp1, Fp2).**

might believe this is not high, it is a reminder that blink/non-blink identification is made in real-time in this study. The second finding is that the variation of the buffer size seems to impact accuracy. In other words, increasing the number of windows in the buffer seems to affect the overall denoising capability of the *onEEGwaveLAD* pipeline. This suggests that future replications of this study with a higher buffer size (>20) can further improve the identification of blinks and non-blinks. Thirdly, the expansion step does not seem to affect accuracy. A marginal effect is visible but insignificant in the second raw of distributions (with $B_s = 15$).

A more detailed analysis of the signal-to-noise ratios (SNR) probability distributions revealed more prominent and visible effects. Fig 5 depicts the probability distributions of the means of the SNR scores obtained from all the participants for all the EEG windows grouped by prediction category (TP, TN, FP, FN) before and after applying the *onEEGwaveLAD* pipeline instantiated with the parameters of Table 1. The whole analytics for the various instantiations across the various parameters of Table 1 can be found in the S1 Appendix (Figs 7–12 in S1 Appendix). In all the cases, the signal-to-noise ratios of the EEG windows in which a blink was detected (TP, orange distributions, first column) were higher than those computed for the same windows in the original signal (blue distributions). This is also confirmed by the Jensen-Shannon divergence between these two probability distributions that is consistent across the

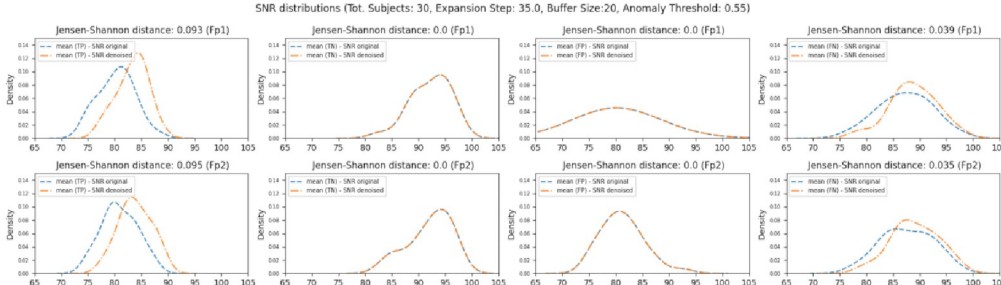

**Fig 5.** Probability distributions of the means of the signal-to-noise ratios (SNR) for all the EEG windows processed for all the subjects (30), grouped by prediction category (TP = True Positives, TN = True Negatives, FP = False Positives, FN = False Negatives) for a specific instantiation of the *onEEGwaveLAD* pipeline (EEG Window Length = 1000ms, Sampling rate = 1024, Mother wavelet = Sym4, Buffer capacity = 20, IF sub-sampling size = 512, Number of IF trees = 100, Anomaly Threshold:0.55, Expansion step = 35).

various instantiations of the *onEEGwaveLAD* pipeline for the true positives. Additionally, by examining the variation of such divergences, contrary to the previous results, it is now possible to see that it increases at higher expansion steps (Figs 7–12 in S1 Appendix). As expected, such divergence is null for the true negatives (TN, second column), showing how the two density distributions of signal-to-noise ratios are identical. This means that in those EEG windows where no blink exists, the *onEEGwaveLAD* correctly did not detect any blink (TN). Thus, no modification of the underlying signal was performed. A similar case is observable for the SNRs of the density distributions associated with the false positives (FP, third column). The *onEEG-waveLAD* could not detect a blink; thus, the underlying signal was not modified. Eventually, an improvement of the SNRs for the false negatives (FN, 4th column) is observable, although it is smaller in magnitude than that associated with the TPs. This is confirmed by the Jensen-Shannon divergence, which is constant across the distributions associated with the various expansion steps but always roughly three times weaker than those observed for the TPs. However, for them, a higher expansion step did not impact increasing SNRs.

Figs 13 to 18 in S1 Appendix present some of the comparisons between the original signal and its denoised counterpart for some of the windows composing the overall EEG signal that was processed by the *onEEGwaveLAD* pipeline, grouped by prediction category (TP, TN, FP, FN). A dashed, tick, vertical red line indicates the presence of a ground truth, plausible blink, identified by the offline algorithm S1 Appendix. The straight, vertical, thinner lines indicated the timesteps at which the *onEEGwaveLAD* pipeline found anomalies, specifically in its phase E (section 3.5).

Fig 13 in S1 Appendix depicts four illustrative cases of true positives (correctly identified blink). In the top EEG window (Win 26), it is possible to see how *onEEGwaveLAD* identified the area around a blink as anomalous. The original signal appears to be correctly denoised within this area by almost half its magnitude. Noticeably, the denoised signal differs from the original even outside this area. This is due to the denoising strategy applying the expansion step, which expanded the area of the identified anomalous vectors at different time steps before and after the blink. Additionally, it is possible to see how the rest of the signal outside this expanded area was correctly not modified. In the second EEG window (Win 30), it is possible to see how only the area before the identified blink was considered anomalous and minimally denoised. This might be explained by the fact that the first part of the blink, before its positive peak, is more transient and thus anomalous than the area after it. In the third EEG window (Win 31), the blink was identified at its end and reasonably denoised. Its magnitude was reduced in some parts while it increased in others. This effect is probably due to the wavelet theory [55] and its motivation that the wavelet coefficients at a particular time step, and the activated frequencies around it, are similar in magnitude and somehow influence each other in the computation of the inverse Discrete Wavelet transform for bringing the signal back in the time domain. Although every coefficient in each *n*-dimensional anomalous vector at different frequencies was reduced, some were minimised with different strengths, as described in phase F of the pipeline (section 3.6), producing such an overall denoised EEG window. The bottom plot (Win 52) showcases how a few anomalous vectors were only identified before the location of a blink peak, and therefore, the underlying signal was minimally denoised. This might have happened because only a few portions of EEG activity before the blink peak were transient and considered abnormal.

Fig 14 in S1 Appendix presents four other EEG windows associated with true positives. The top plot (Win 69) was identified as a true positive, but none of the abnormal vectors were in the area surrounding the blink peak. This suggests that using the selected mother wavelet (Sym4) is probably not always ideal for representing the characteristics of a blink. Despite this, the *onEEGwaveLAD* pipeline was able to detect and mitigate some transient activity. In the

second EEG window (Win 116) and fourth (Win 302), it is possible to see how only a few identified anomalous vectors lead the underlying signal to be deemed a true positive. In this case, the *onEEGwaveLAD* could not identify anomalous transient activity around the blink peak; thus, no modification was made. This might be explained by the length of such blinks (roughly 400 ms), which are slightly longer than the others. Therefore, the signal around it is less variable over time, resulting in a lack of transient activity that can be considered anomalous. The third EEG window (Win 132) highlights how anomalous vectors were identified and denoised around the blink peak. However, other abnormal activity was identified at the end of the EEG window, even if no ground truth blink exists. This suggests that the mother wavelet (Sym 4) might not be fully appropriate to mimic the characteristics of a blink. Fig 15 in S1 Appendix illustrates four examples of true negatives. Here, the *onEEGwaveLAD* pipeline was successfully able to ignore transient activity and, therefore, leave the underlying EEG signals intact. Note also that, in the bottom EEG window (Win 24), the area surrounding the peak around 500ms could resemble a blink. However, the offline blink detection algorithm did not deem this a blink, probably because of its frequency domain characteristics or topographical activation in space. Successfully, *onEEGwaveLAD* did not identify any abnormal activity in the DWT coefficients at the various scales (frequencies), leaving the EEG signal intact.

Fig 16 in S1 Appendix presents false positive EEG windows, whereby the offline detection algorithm spotted a true blink, but the *onEEGwaveLAD* pipeline could not identify it; thus, no modification of the EEG activity was made. In the top case (Win 21), the *onEEGwaveLAD* pipeline did not spot the blink. Either the information in the moving buffer was insufficient to allow for successful discrimination of anomalous vectors, or it was highly variable, thus preventing the Isolation Forest algorithm from successfully learning the characteristics of anomalies. In the second plot (Win 56), the blink happened right at the beginning of the EEG window. Again, unsuccessfully, the *onEEGwaveLAD* was not able to spot the blink, even if, in the computation of the DWT coefficients, the previous window was considered to avoid the effect of the cone of influence and mitigate edge effects, as described in section 3.2. However, in the bottom case (Win 336), the blink occurs at the end of the window, and in this case, edge effects might be present, as no data exists on its right side. This might explain the incapability of the *onEEGwaveLAD* pipeline to detect a blink in such special circumstances. In the third plot (Win 252), the *onEEGwaveLAD* pipeline could not detect such blinks. This might be explained by a special case in which two consecutive blinks at the beginning of the window exist, thus altering the normal shape of a blink that was not identifiable by the detection strategy of the pipeline using the selected mother wavelet (Sym4).

Eventually, Figs 17 and 18 in S1 Appendix depict examples of false negatives. Here, the *onEEGwaveLAD* detected some transient activity and performed denoising around the areas identified as abnormal. In the top of Fig 17 in S1 Appendix (Win 34) and in the bottom of Fig 18 in S1 Appendix (Win 236), just a minimal amount of anomalies have been found, but enough to be characterised as false negatives, since at least one anomaly in each of them was necessary. Clearly, if the minimum requirement of one anomaly to classify an EEG window as containing a blink is extended to more than 1, these are not false negatives but true negatives. Contrarily, the third and fourth of Fig 17 in S1 Appendix (Wins 131, 139) and the second of Fig 18 in S1 Appendix (Wins 208) are certainly false negatives, as many anomalies were found, but no blink was present. In these cases, it is evident how the selected mother wavelet (Sym 4) is not always ideal for only trying to find blinks in a signal but also for other types of transient activity in EEG signals. Another explanation might be how a current EEG window is examined and evaluated, given the past information in the preceding moving buffer. Suppose the data in such a buffer is highly condensed, and the new window is highly variable. In that case, many points in such a new window might be considered anomalies by the *onEEGwaveLAD* pipeline.

Eventually, the second plot of Fig 17 in S1 Appendix (Win 58) and the first and third of Fig 18 in S1 Appendix (Wins 174, 230) present an erroneous behaviour toward the end of the EEG window. Here, anomalies are found in the DWT coefficient space, but probably because of the problem of the cone of influence, when such coefficients are manipulated in magnitude, and reconverted in the time domain with the inverse Discrete Wavelet Transform, then the underlying EEG signal is reconstructed wrongly.

According to the above critical discussion, a synthesis of findings is schematically presented below:

- The manipulation of the buffer size revealed a marginal impact on the denoising capacity of the *onEEGwaveLAD* pipeline when analysed between subjects; however, when examined within the subjects, it appeared to deliver an increasing impact on denoising EEG windows. This suggests that larger sizes can further enhance the capability of the pipeline to better learn from past EEG data and identify anomalous transient activity in it. A note of caution is that the buffer size cannot be very large because the resulting training of the anomaly detection model (Isolation Forest) can take longer than the length of the EEG window, thus, in practice, destroying the real-time nature of the *onEEGwaveLAD* pipeline.

- The manipulation of the expansion step revealed no impact on denoising capability when examined between-subject; however, when investigated within-subject, it demonstrated an effect for denoising the EEG transient anomalous activity around a blink in the true positive cases. In particular, abnormal EEG segments were often found in the area before a blink peak, probably more distinct and transient from the overall EEG data than its right part.

- the selected mother wavelet (Sym 4), although it is probably the closest function to resemble the characteristic of a blink, among other symmetrical functions, was not always an ideal approach for the identification of only blinks since it had an effect on denoising capability in the EEG windows associated to false negatives, that means those that did not contain a blink. The selected mother wavelet might have been sensitive to other types of artefacts not necessarily associated with blinks, as it might include saccades or muscle movements. However, note that this is intuitive speculation since it has not yet been tested and is part of future work. In other words, different instantiations of the pipeline are necessary, using different mother wavelets resembling other artefact types, including saccades and muscle movements among others.

- the *onEEGwaveLAD* pipeline demonstrated perfect accuracy on the true negatives since no modification of the underlying data was correctly performed.

- As intuitively expected, the *onEEGwaveLAD* pipeline showed some problems denoising EEG data at the end of an EEG window. This is likely because of the predicted boundary effects on the right side of an EEG window, which are linked to the theoretical problem of the cone of influence associated with the discrete wavelet transformation.

- the *onEEGwaveLAD* pipeline, with the selected mother wavelet, exhibited some issues in identifying two consecutive blinks in the same EEG window. This is likely due to the waveform of such consecutive blinks, which is different from that of only one, thus deviating from the shape of the selected mother wavelet.

- the selection of the value of 1 used to categorise an EEG window as a true positive, which means containing a blink, is not the most effective approach. Extending such value can lead to a further grid search, but there is the problem of establishing the upper bound, suggesting that alternative methods should be envisioned.

## 6 Conclusions

The problem of artefact identification and reduction in EEG signals is significant for facilitating the understanding of brain responses in clinical applications and brain-computer interfaces. While many offline methods have been devised in the literature, demonstrating incremental capacity for denoising EEG data, this is not the case for online methods, which are increasingly invoked for real-time artefact reduction. Offline methods can take advantage of the entire length of recorded multi-channel EEG data, successfully learn the characteristics of artefacts, statistically or not, and can be supervised by humans. In contrast, online methods can only exploit EEG data up to the time recorded and cannot use human supervision since they need to work in real time. To bridge this gap, this research study contributes to the body of knowledge by proposing *onEEGwaveLAD*, a novel fully automated online EEG wavelet-based learning adaptive denoising framework for artefact identification and mitigation. It was demonstrated how such a framework can be instantiated for the particular use case of blink identification and reduction. Results are promising, demonstrating how such instantiation could detect blinks in real time for the Fp1 and Fp2 channels and increase the signal-to-noise ratio in the time interval around them while preserving it on the rest of the EEG data. Many possible future improvements of such a framework are envisioned. Firstly, *onEEGwaveLAD* can be instantiated and tested to identify and reduce other types of ocular artefacts, such as saccades. Similarly, it can be adopted for identifying and removing muscle artefacts and more difficult-to-spot noise forms, such as cardiac artefacts. Dedicated studies can focus on specific investigations of the impact of the moving buffer size, the number of trees, or the sub-sampling size of the Isolation Forest algorithm. Similarly, a detailed analysis of the impact of the anomaly threshold or the expansion step on the resulting denoising capacity can be conducted. Future work can also include the development of an ad hoc symmetric mother wavelet to approximate better the characteristics and ideal shape of a blink and other types of artefacts. Focusing on only some frequency bands rather than all could improve the denoising strategy of the DWT coefficients. Eventually, smaller windows of EEG data can be considered, improving the real-time nature of *onEEGwaveLAD* and thus opening new paths for developing artefact-free online brain-computer interfaces in ecological settings.

## Supporting information

**S1 Appendix.**
(PDF)

## Author Contributions

**Conceptualization:** Luca Longo.

**Data curation:** Luca Longo.

**Formal analysis:** Luca Longo.

**Funding acquisition:** Richard B. Reilly.

**Investigation:** Luca Longo.

**Methodology:** Luca Longo.

**Project administration:** Luca Longo.

**Resources:** Luca Longo.

**Software:** Luca Longo.

**Supervision:** Richard B. Reilly.

**Validation:** Luca Longo.

**Visualization:** Luca Longo.

**Writing – original draft:** Luca Longo.

**Writing – review & editing:** Luca Longo, Richard B. Reilly.

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
