## [Decision Letter · Decision Letter 0]

20 Sep 2024

PONE-D-24-35626onEEGwaveLAD: A fully automated online EEG wavelet-based Learning Adaptive DenoiserPLOS ONE

Dear Dr. Longo,

Thank you for submitting your manuscript to PLOS ONE. After careful consideration, we feel that it has merit but does not fully meet PLOS ONE’s publication criteria as it currently stands. Therefore, we invite you to submit a revised version of the manuscript that addresses the points raised during the review process.

We look forward to receiving your revised manuscript.

Kind regards,

Jerritta Selvaraj

Academic Editor

PLOS ONE

Journal Requirements:

Reviewers' comments:

Reviewer's Responses to Questions

**Comments to the Author**

1. Is the manuscript technically sound, and do the data support the conclusions?

Reviewer #1: Yes

Reviewer #2: Yes

2. Has the statistical analysis been performed appropriately and rigorously? 

Reviewer #1: Yes

Reviewer #2: Yes

3. Have the authors made all data underlying the findings in their manuscript fully available?

Reviewer #1: Yes

Reviewer #2: Yes

4. Is the manuscript presented in an intelligible fashion and written in standard English?

Reviewer #1: Yes

Reviewer #2: Yes

5. Review Comments to the Author

Reviewer #1: 1. Keywords should be limited to maximum of 6.

2. Justified the paragraph

2. Abstract should be more concise

3. All the formulas should be numbered (Some formula numbers are missing eg: Page no. 7)

4. Add recent article in the reference

Reviewer #2: Q1. Why was sym4 mother wavelet chosen among other wavelets for noise reduction in EEG Signal?

Q2. What was the duration of the EEG data for the 37 participants from the N170 dataset of ERP CORE form and how does the other artefacts such as muscle movements impact the signal?

Q3. It is mentioned in the manuscript that the anomalies in this algorithm is assumption of specific data points in cardinality and different from other points-Justify

Q4. Does all the data from dataset is included as random dimension values rd1 to rdc ?

Q4. Figure 2,3 image is not clear , can use a HD image.

6. PLOS authors have the option to publish the peer review history of their article (what does this mean?). If published, this will include your full peer review and any attached files.

Reviewer #1: No

Reviewer #2: No

---

## [Author Response · Author response to Decision Letter 0]

25 Sep 2024

Q1. Why was sym4 mother wavelet chosen among other wavelets for noise reduction in EEG Signal?

PAGE 13, SECTION 4.2 (IN RED OF RESUBMITTED MANUSCRIPT)

"This is chosen because its shape highly resembles an ocular eye 514

blink [61] (fig.2, middle) with a positive peak. "

Q2. What was the duration of the EEG data for the 37 participants from the N170 dataset of ERP CORE form and how does the other artefacts such as muscle movements impact the signal?

PAGE 13, SECTION 4.1 (IN RED OF RESUBMITTED MANUSCRIPT)

"The recording length averages 581 seconds with a 484

standard deviation of 55 seconds across participants."

PAGE 20, SECTION 5 (IN RED OF RESUBMITTED MANUSCRIPT)

"The selected mother wavelet might have been sensitive to other types of artefacts not necessarily associated with

blinks, as it might include saccades or muscle movements. However, note that this

is intuitive speculation since it has not yet been tested and is part of future work.

In other words, different instantiations of the pipeline are necessary, using

different mother wavelets resembling other artefact types, including saccades and

muscle movements among others."

Q3. It is mentioned in the manuscript that the anomalies in this algorithm is assumption of specific data points in cardinality and different from other points-Justify

PAGES 7 AND 8, SECTION 3,4 (IN RED OF RESUBMITTED MANUSCRIPT)

" In other words, anomalies are rare instances of a dataset, and their

characteristics in an n-dimensional space should be drastically different from those of the other instances and, therefore, easily identifiable."

Q4. Does all the data from dataset is included as random dimension values rd1 to rdc ?

YES, THE ENTIRITY OF THE DATA, FOR EACH PARTICIPANT, WAS CONSIDERED

Q5. Figure 2,3 image is not clear , can use a HD image. 

IMAGES HAVE BEEN RE-UPLOADED WITH AN HIGHER QUALITY.

IF ACCEPTED, PLOS ONE HAS ITS OWN WAY OF DEALING WITH IMAGES, AND THIS WILL BE STRICLY FOLLOWED IN THE LAST STEP OF SUBMISSION.

---

## [Decision Letter · Decision Letter 1]

18 Oct 2024

onEEGwaveLAD: A fully automated online EEG wavelet-based Learning Adaptive Denoiser for artefacts identification and mitigation

PONE-D-24-35626R1

Dear Dr. Longo,

We’re pleased to inform you that your manuscript has been judged scientifically suitable for publication and will be formally accepted for publication once it meets all outstanding technical requirements.

Kind regards,

Jerritta Selvaraj

Academic Editor

PLOS ONE

Additional Editor Comments (optional):

Reviewers' comments:

Reviewer's Responses to Questions

**Comments to the Author**

1. If the authors have adequately addressed your comments raised in a previous round of review and you feel that this manuscript is now acceptable for publication, you may indicate that here to bypass the “Comments to the Author” section, enter your conflict of interest statement in the “Confidential to Editor” section, and submit your "Accept" recommendation.

Reviewer #1: All comments have been addressed

Reviewer #2: All comments have been addressed

2. Is the manuscript technically sound, and do the data support the conclusions?

Reviewer #1: Yes

Reviewer #2: Yes

3. Has the statistical analysis been performed appropriately and rigorously? 

Reviewer #1: Yes

Reviewer #2: I Don't Know

4. Have the authors made all data underlying the findings in their manuscript fully available?

Reviewer #1: Yes

Reviewer #2: Yes

5. Is the manuscript presented in an intelligible fashion and written in standard English?

Reviewer #1: Yes

Reviewer #2: Yes

6. Review Comments to the Author

Reviewer #1: I would to like to appreciate the author for delivering a good research paper. There are some minor corrections to be included in the paper as follows:

1. Reduce the keywords to count of 6.

2. Justify the paragraph.

3. Sub figures are not numbered and not clearly explained.

Update the above changes.

Reviewer #2: (No Response)

7. PLOS authors have the option to publish the peer review history of their article (what does this mean?). If published, this will include your full peer review and any attached files.

Reviewer #1: No

Reviewer #2: No

---

## [Editor Report · Acceptance letter]

23 Oct 2024

PONE-D-24-35626R1 

PLOS ONE

Dear Dr. Longo, 

I'm pleased to inform you that your manuscript has been deemed suitable for publication in PLOS ONE. Congratulations! Your manuscript is now being handed over to our production team.

Kind regards, 

on behalf of

Dr. Jerritta Selvaraj 

Academic Editor

PLOS ONE